# Analysis of Non-Idealities on CMOS Passive Mixers

**Antonio D. Martinez-Perez** [1,*] , **Francisco Aznar** [2] , **Guillermo Royo** [1] **and Santiago Celma** [1]

[1]   Group of Electronic Design (GDE)-Aragon Institute of Engineering Research, Universidad de Zaragoza, 50009 Zaragoza, Spain; royo@unizar.es (G.R.); scelma@unizar.es (S.C.)

[2]   Group of Electronic Design (GDE)-Aragon Institute of Engineering Research, Centro Universitario de la Defensa, 50090 Zaragoza, Spain; faznar@unizar.es

\*   Correspondence: adimar@unizar.es

**Abstract:** In the current state of the art, WiFi-alike standards require achieving a high Image Rejection Ratio (IRR) while having low power consumption. Thus, quadrature structures based on passive ring mixers offer an attractive and widely used solution, as they can achieve a high IRR while being a passive block. However, it is not easy for the designer to know when a simple quadrature scheme is enough and when they should aim for a double quadrature structure approach, as the latter can improve the performance at the cost of requiring more area and complexity. This study focuses on the IRR, which crucially depends on the symmetry between the I and Q branches. Non-idealities (component mismatches, parasitics, etc.) will degrade the ideal balance by affecting the mixer and/or following/previous stages. This paper analyses the effect of imbalances, providing the constraints for obtaining a 40 dB IRR in the case of a conversion from a one-hundred-megahertz signal to the five-gigahertz range (upconversion) and vice versa (downconversion) for simple and double quadrature schemes. All simulations were carried out with complete device models from 65 nm standard CMOS technology and also a post-layout Monte Carlo analysis was included for mismatch analysis. The final section includes guidelines to help designers choose the most adequate scheme for each case.

**Keywords:** CMOS technology; image rejection ratio; mismatch impact; quadrature mixer





## 1. Introduction

Almost all modern communication systems require a mixer for frequency conversion, as, currently, most transceivers are heterodyne. This means that they process the signal at an intermediate frequency (IF), but they transmit and/or receive in a higher frequency range, indicated as the radio frequency (RF) band. Thus, the system must shift the signal frequency range before transmitting and after receiving. To carry out this operation, a local oscillator (LO) reference signal is also required.

Indeed, transceiver implementations usually employ quadrature architectures based on multiple mixers to avoid transmitting or receiving an image signal. This issue in transmission means that the system will not only transmit the desired signal in the selected channel but also an undesirable copy of the signal on another band out of the channel, causing, therefore, interference on that band. On the other hand, signals present in the reception image band will be added as noise to the received signal [1].

For this reason, physical layer specifications establish the limits of an admissible image for adequate communication by a system using the standard. Typically, the most critical limitations are related to the image rejection ratio (IRR), i.e., the relationship between the power of the desired signal and the power of the image [2]. Due to the importance of the parameter, it is also a figure of merit for the elements involved in the frequency conversion as all of the architecture contributes to the image cancellation [3].

This work is contextualized in the development of a CMOS integrated transceiver for remote antenna unit (RAU) applications. In it, a one-hundred-megahertz IF signal from an optical link is converted to a five-gigahertz RF band for wireless communication and vice

versa. The IF-over-fiber allows for a very competitive trade-off between the advantages of transmitting the RF or base-band signal over fiber in terms of cost per unit and global cost [4], while the RF range enables communication in a WiFi-alike standard. To cope with reference standard specifications, the mixer's design must guarantee a minimum of a 40 dB IRR [2].

This study focuses on passive mixers, which are widely used in the current state of the art for integrated communication systems in nanometer CMOS processes. Despite their incapability to provide gain, their performance in terms of figure noise, linearity, and power consumption exceeds those of the active alternatives [5]. The following conversion schemes are based on a MOS passive ring mixer (see Figure 1), implemented in nanometer CMOS technology.

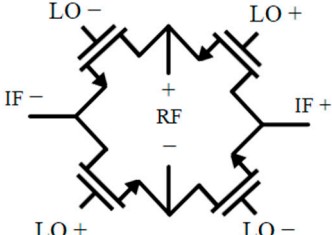

**Figure 1.** Schematic representation of the MOS passive ring mixer at the transistor level. Note that all signals from block-level schemes in this work are differential.

Parameters such as linearity [6] and noise [7], and even mixer impedances [8], have been analyzed in detail in the literature for mixers; nevertheless, an imperfect balance complicates significantly the analysis of this kind of system. Thus, previous studies [3] do not delve into the relationship between an imbalance and the IRR, providing qualitative explanations instead and restricting the analysis to certain non-idealities. However, this paper aims to provide quantitative results for the relationship between any system imbalance and IRR degradation and identify the most critical point to achieve a high IRR. This work covers simple and double quadrature schemes in upconversion and downconversion operation. Additionally, we not only evaluate the imbalance in the mixer stage and the input signal but also in contiguous stages, in the form of a mismatch between the equivalent source and load impedance, as the importance of load and source effects on a mixer's general parameters is known [9]. These effects, far from being negligible, can be critical to attaining high IRR requirements. Thus, the conclusions of this study may be of great interest to analogue RF designers, especially those working in standards with high IRR requirements.

This work is outlined as follows. Section 2 presents a brief description of the frequency conversion operation and an adequate interpretation of the IRR as a figure of merit; the mixer's topology and its specifications are also described. Section 3 analyses the impact of mismatch and quadrature errors on the IRR for both upconversion and downconversion cases. Additionally, both cases are evaluated by a Monte Carlo analysis to provide a complete statistical approach. Section 4 discusses the results, comparing the obtained limits for different cases and schemes. The final section includes the main concluding remarks.

## 2. Frequency Conversion

A signal multiplied by a single tone produces two sidebands located at the sum and difference of input frequencies, respectively. Thus, if the IF and LO signals are the inputs of a mixer carrying out an upconversion, the output will be the signal replicated at $f_{RF1} = f_{LO} + f_{IF}$ and $f_{RF2} = f_{LO} - f_{IF}$. Similarly, in a receiver performing a downconversion, an output at $f_{IF}$ is obtained from $f_{RF1} + f_{LO}$ as well as from $f_{LO} - f_{RF2}$. In both cases, one band ($f_{RF1}$ or $f_{RF2}$) is defined as the desired signal and the other one as the image signal.

In order to reject the image signal, quadrature schemes can be used. A quadrature signal is formed by two components: in-phase (I) and quadrature (Q). Both are equal but ideally displaced 90° from each other. In a simple quadrature scheme, two mixers are used;

both are connected to RF and each one to the I or Q components of IF ($\text{IF}_\text{I}$ or $\text{IF}_\text{Q}$) and LO ($\text{LO}_\text{I}$ or $\text{LO}_\text{Q}$).

Typically, a passive polyphase filter (PPF) is the element that implements the single or differential to quadrature conversion or vice versa [10]. These filters are RC networks that can achieve a certain IRR for a certain desired frequency band [11]. A detailed analysis by the authors about the IRR limitations of these elements can be found in [12].

This means that in a downconversion (Figure 2a), the RF signal is multiplied by $\text{LO}_\text{I}$ and $\text{LO}_\text{Q}$ in mixers to generate $\text{IF}_\text{I}$ and $\text{IF}_\text{Q}$ components, and then a passive polyphase filter converts them in an IF differential signal. In an upconversion (Figure 2b), the IF signal is split by the PPF and each component is multiplied by $\text{LO}_\text{I}$ or $\text{LO}_\text{Q}$. These components are added to produce a RF signal.

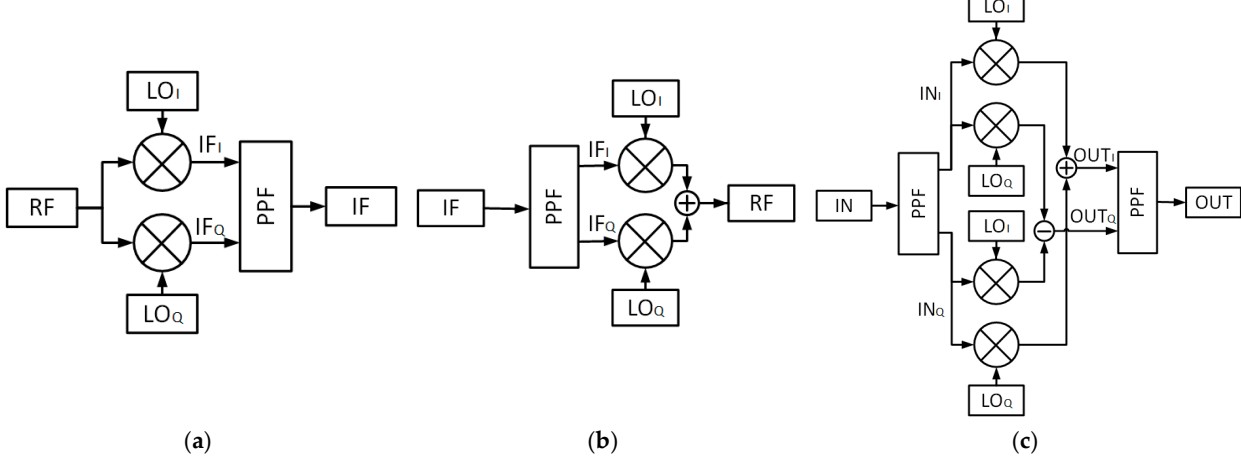

(a)          (b)          (c)

**Figure 2.** Block diagrams of analyzed schemes: (**a**) Simple quadrature downconversion. (**b**) Simple quadrature upconversion. (**c**) Double quadrature, in which the same scheme can be used for upconversion or downconversion. It should be noticed that all signals are differential.

On the other hand, in a double quadrature scheme (Figure 2c), four mixers are employed. A PPF decomposes the input signal, either RF (downconversion) or IF (upconversion), in quadrature form and each component is multiplied by $\text{LO}_\text{I}$ and $\text{LO}_\text{Q}$. The outputs are combined as $\text{OUT}_\text{I} = \text{IN}_\text{I} \cdot \text{LO}_\text{I} + \text{IN}_\text{Q} \cdot \text{LO}_\text{Q}$ and $\text{OUT}_\text{Q} = \text{IN}_\text{I} \cdot \text{LO}_\text{Q} - \text{IN}_\text{Q} \cdot \text{LO}_\text{I}$ and, finally, another PPF recomposes the quadrature signal to differential mode. This more complex architecture virtually relaxes the requirements on the components at the cost of requiring more elements.

Under ideal conditions, both single and double quadrature architectures cancel the image signal. However, the presence of non-idealities, such as device mismatches or imbalanced I and Q signals, leads to an imperfect cancellation [6]. These effects can be modeled as an error in the phase and/or amplitude in one input quadrature, while the rest of the system, including the other input, remains under ideal conditions [3]. In the single quadrature downconversion, the error must be included in the LO signal, whereas the other cases can insert it into the LO signal as well as the other input. The following expression relates the image rejection ratio with the input quadrature imbalance [10]:

$$\text{IRR} = \frac{1 + 2\text{A}_\text{BAL} \cos \Delta\theta + \text{A}_\text{BAL}^2}{1 - 2\text{A}_\text{BAL} \cos \Delta\theta + \text{A}_\text{BAL}^2} \tag{1}$$

where $\text{A}_\text{BAL}$ is the amplitude balance, i.e., $\text{A}_\text{BAL} = |\text{I}| / |\text{Q}|$ and $\Delta\theta$ is the phase deviation from the ideal 90° between the I and Q signals.

Thus, the IRR can describe the quality of a quadrature signal as well as, thanks to Equation (1), the severity of mismatches in actual mixers [3]. Indeed, this work expands the idea: IRR degradation can also quantitatively evaluate the performance of any part of

the system or the impact of other non-idealities if all remaining elements are ideal. This approach provides two key advantages: first, the IRR becomes a normalized reference to optimize a discrete part inside of the whole scheme; and, second, the quantification of the impact of the possible issues allows for an easier identification of the main obstacles to attain a high image rejection ratio.

Although the IRR is virtually infinite under ideal balance operation conditions, i.e., assuming ideal quadrature of the input signal [3], balanced source and load impedances [9], and no device mismatches [13], this is not a realistic scenario. In practice, non-idealities of real components, such as parasitics and variations resulting from the fabrication process, will unbalance one or more of these ideal conditions, and hence they will affect the IRR. This degradation only depends on the imbalance regardless of the cause. Thus, this analysis benefits from this induction and studies how deviations from the ideal balance operation conditions affect the IRR to estimate the effect of any non-ideality.

In order to carry out a complete analysis of the IRR, a testbench based on a CMOS passive mixer was implemented with a passive ring mixer topology in 65 nm CMOS technology. Note that the internal mixer's design is the same regardless of whether the circuit is a simple or double quadrature scheme. They are formed by four NMOS transistors with their gates controlled by LO signals and sources and drains to IF or RF (see Figure 1). The transistors operate as switches, being forced to alternate between saturation and cut-off regions. In order to do this, they are polarized close to the threshold voltage. It is important to remark that two and four mixers are needed for the simple and double quadrature schemes, respectively. For this topology, the addition operation can be implemented as a direct connection between the outputs. In the same way, the subtraction operation can be carried out by swapping the differential paths in one of the operators. The simulation testbench includes source and load impedances and parasitic capacitances of 200 fF.

For testing the upconversion operation under demanding conditions, we have intentionally chosen a low value of 100 MHz for the IF signal with a 5 GHz LO signal. This results in very close 5.1 GHz RF and 4.9 GHz image signals. The local oscillator topology used produces differential quadrature signals; thus, it is modeled as two differential sinusoid signals with a spare 90° in phase and an amplitude of 300 mV and 900 mV of common-mode value. The IF signal for upconversion is also a quadrature signal with an amplitude of 100 mV and a common-mode value of 600 mV.

For testing the downconversion operation under the same demanding conditions as those for upconversion, the RF signal has a frequency of 5.1 GHz and an amplitude of 100 mV and a common-mode value of 600 mV, but it is not in quadrature in a single quadrature scheme; it is differential. The LO signal is the same as in the upconversion case. Table 1 presents the main results of the mixer stage from the post-layout simulation.

**Table 1.** Single Quadrature Post-Layout Specifications.

| Parameter | Upconversion | Downconversion |
|---|---|---|
| Conversion Losses | 14 dB | 2 dB |
| P1dB | 0 dBm | −8.8 dBm |
| IIP3 | 1 dBm | 2.6 dBm |
| Input–Output Isolation | 85 dB | 100 dB |
| LO–Output Isolation | 103 dB | 103 dB |
| Input Frequency | 100 MHz | 5.1 GHz |
| Output Frequency | 5.1 GHz | 100 MHz |
| $W_{NMOS}$ | 32 μm | 32 μm |
| $L_{NMOS}$ | 140 nm | 140 nm |

## 3. Results

This study covered both topologies, simple quadrature (SQ) and double quadrature (DQ), under the scenarios of upconversion and downconversion operations. In each of these cases, the analysis evaluates the following imbalances: input signal quadrature

deviation (phase and/or amplitude); source or load impedance mismatch; and mismatch on the mixer stage transistor, calculating the worst mismatch combination among all transistors of the scheme. Additionally, a statistical analysis by the Monte Carlo method was carried out to complement the simulation data.

### *3.1. IRR Analysis in Upconversion*

In upconversion, the IF signal must be decomposed in $IF_I$ and $IF_Q$. However, RF, after the mixers, would be obtained as a differential or quadrature signal depending on the scheme. This implies that a double quadrature setup requires an additional PPF tuned at the RF signal frequency to recompose RF from $RF_I$ and $RF_Q$. Furthermore, the double quadrature scheme (Figure 2c) uses twice as many mixer stages as the simple scheme (Figure 2a).

### 3.1.1. Input Quadrature

The quality of input quadrature signals (IF and LO) limits the maximum IRR reachable at the output, as shown in Figure 3a. In a simple quadrature, the IF phase error $\Delta\theta$ and the amplitude balance $A_{BAL}$ impact the IRR according to (1). Single quadrature LO deviations produce a similar effect, although it seems to relax the $A_{BAL}$ constraints at the cost of a more restrictive phase requirement. On the other hand, only deviations in IF affect the double quadrature scheme. The architecture cancels the error in one of the quadrature inputs if the other has a perfect quadrature. Either of the inputs can benefit from these advantages, but the LO frequency is much higher than the IF frequency and, as a consequence, the benefits are greater. The IF deviation impact results are quite similar to the single quadrature case, although amplitude errors impose a tougher penalty.

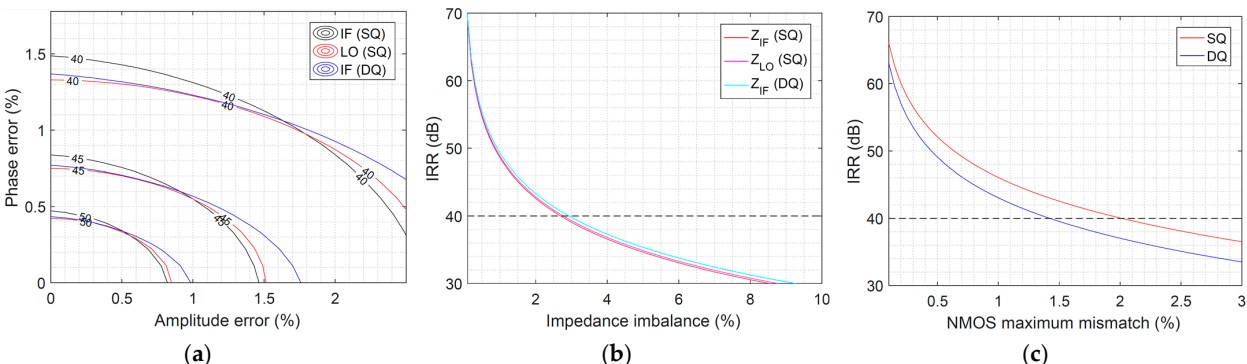

**Figure 3.** IRR curves in upconversion for simple quadrature (SQ) and double quadrature (DQ) schemes: (**a**) Imperfect input quadrature. LO amplitude and phase errors are self-corrected in the double quadrature scheme; hence, their curves are not drawn. (**b**) Impact of impedance imbalance. Mismatches in $Z_{RF}$ and $Z_{LO}$ (DQ) are negligible by comparison. (**c**) Worst mismatch scenario in the mixer's NMOS W/L ratio.

### 3.1.2. Impedance Imbalance

Source ($Z_{LO}$ and $Z_{IF}$) and load ($Z_{RF}$) impedance imbalances are also important issues, as shown in Figure 3b. In a simple quadrature scheme, the IRR is worsened when impedances in the IF and LO ports suffer from a mismatch. They suppose a quite similar impact and a mismatch no larger than 2.75% should be guaranteed for both to achieve 40 dB. The load impedance does not affect the IRR because the signal has already been converted to differential mode.

On the other hand, in a double quadrature approach, the mismatch restriction in the LO port is notably relaxed to the point that it becomes negligible in comparison with the others.

A mismatch in IF impedance is slightly less detrimental than in the simple quadrature case: a 3% mismatch for a 40 dB IRR. It is remarkable that, despite having a quadrature output, the effect of the load impedance mismatch is virtually cancelled.

### 3.1.3. Mixer Mismatch

This analysis compares the robustness of both schemes against a mismatch on the transistors. All transistors receive the same deviation, but they can differ in the sign. Then, the scheme is evaluated in the worst case, i.e., the combination of deviations that produces the largest IRR degradation. Specifically, worst cases are opposite variations in each single quadrature mixer and mixers with the same IF input for double quadrature mixers. The comparison in Figure 3c reveals that the single quadrature scheme is more reliable than the double quadrature scheme for this kind of non-ideality. Quantitatively, there is a 3 dB IRR difference between both schemes, which translates into a mismatch requirement of 1.4% maximum deviation for the double quadrature scheme to attain a 40 dB IRR, whereas the simple quadrature scheme can relax the constraint to 2%.

### 3.2. IRR Analysis in Downconversion

In contrast to upconversion, in downconversion, IF must be recomposed from $IF_I$ and $IF_Q$. The RF input signal is required in a differential (simple quadrature) or quadrature (double quadrature) topology. In a double quadrature topology, an additional PPF is needed to generate the quadrature of RF.

### 3.2.1. Input Quadrature

Opposite to upconversion, the quality of the signal input (RF) does not limit the achievable IRR. In a simple quadrature scheme, RF is a differential signal, while in a double quadrature scheme, RF quadrature errors are corrected if the LO signal is ideal. Similarly, LO quadrature errors are cancelled in the double quadrature scheme as long as the RF signal is ideal. The image in double quadrature downconversion is the result of errors in both inputs. Due to the dependence on both signal errors, their effects on the IRR can be neglected in comparison with the impact of other imperfections.

However, the LO phase error and amplitude imbalance impact on the IRR in the single quadrature scheme follows Equation (1) as shown in Figure 4a.

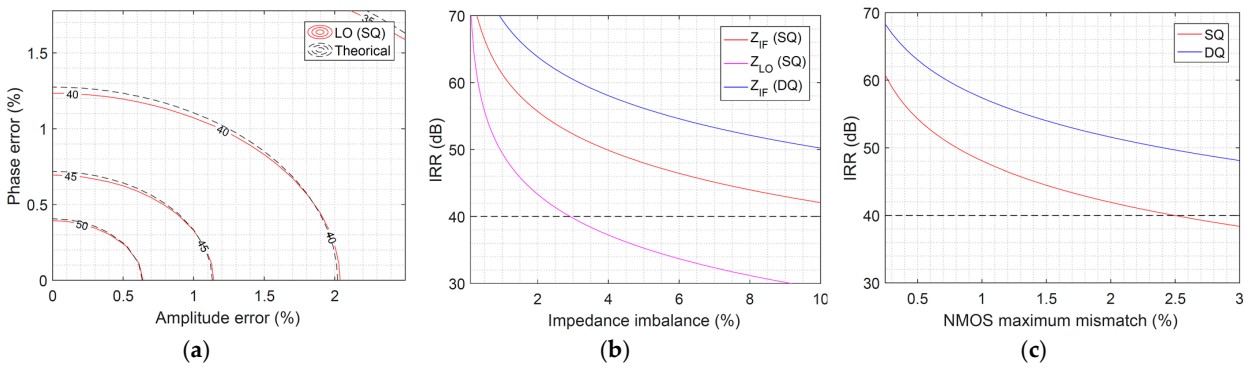

(a)    (b)    (c)

**Figure 4.** IRR curves in downconversion for simple quadrature (SQ) and double quadrature (DQ) schemes: (**a**) Imperfect input quadrature. RF does not affect the IRR as it is differential for SQ and its error results are cancelled by the ideal LO in DQ. Additionally, DQ cancels the LO error if the RF quadrature is ideal. Theoretical results from Equation (1) are included as a reference (dashed line). (**b**) Impact of an impedance imbalance. Mismatches in ZRF and ZLO (DQ) are corrected in the schemes. (**c**) Worst mismatch scenario in the mixer's NMOS W/L ratio.

### 3.2.2. Impedance Imbalance

In downconversion, an impedance imbalance in the RF port can be neglected in the same way the quality of the input quadrature for that port is: for a simple quadrature scheme, the mismatch between differential impedances does not alter the IRR; and in a double quadrature scheme, the effect of RF source impedances is cancelled if the LO signal is ideal. Additionally, the LO impedance error is cancelled in a double quadrature scheme if the IF signal has the ideal balance.

However, in the single quadrature scheme, a LO impedance mismatch has a strong impact and should be under 3% to obtain an IRR above 40 dB, as shown in Figure 4b. Nevertheless, a double quadrature setup does increase the performance against an IF load mismatch, although in a simple quadrature scheme it is not critical.

### 3.2.3. Mixer Mismatch

Similarly to the upconversion case, the analysis evaluates a mismatch under the worst combination. The same worst mismatch situation explained in Section 3.1.3 was evaluated. However, contrary to upconversion, the double quadrature approach provides an improvement regarding the simple quadrature scheme. Nevertheless, downconversion has notably more relaxed constraints than upconversion: single and double quadrature schemes can tolerate 2.5% and 5% deviations, respectively.

Additionally, it should be noted that the worst combination for the double quadrature scheme is less probable than the one for the single quadrature scheme due to the higher number of elements. Considering that all deviations are equal but can be different in sign, it would be 1/16 of the cases versus 1/8.

### 3.3. Statistical IRR Analysis

In previous subsections, the worst case of transistor mismatch was evaluated for both schemes: single and double quadrature. However, the likelihood of that case depends on the number of transistors, which differs from one scheme to another. Thus, a statistical approach, in the form of Monte Carlo analysis, is convenient for a fair comparison. Histograms from Figure 5 are the result of a 1000 sample Monte Carlo analysis employing the statistical technology model for the mixer transistor.

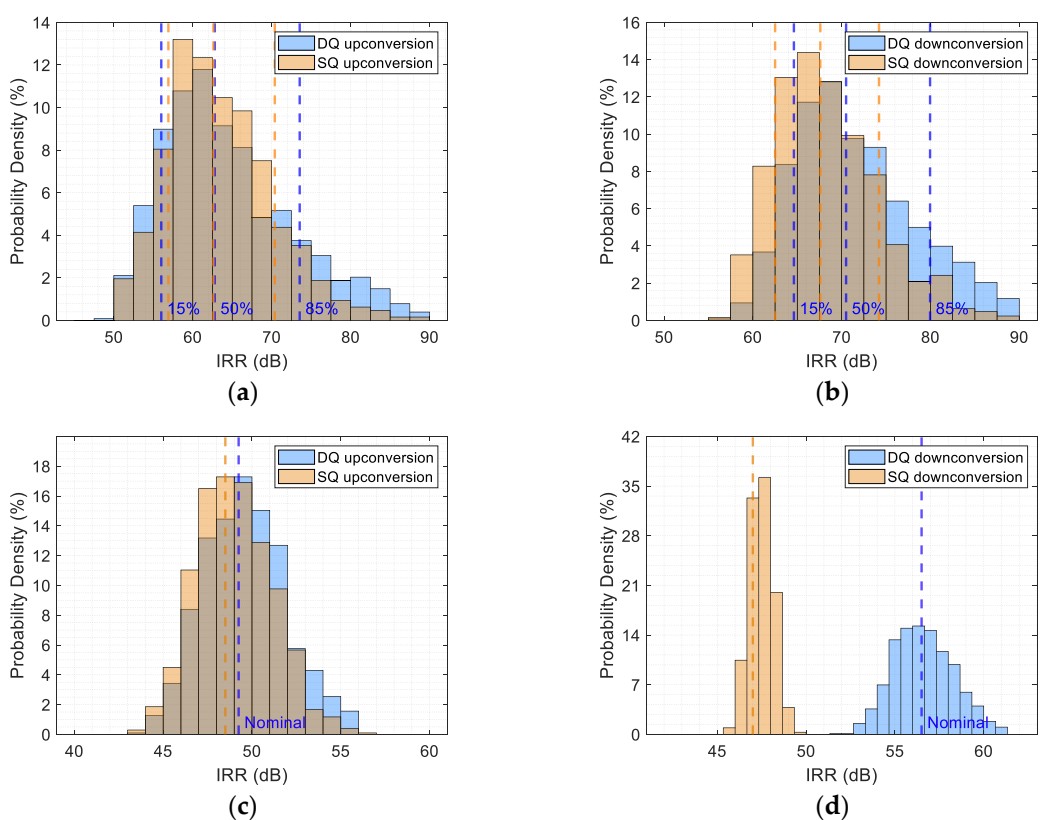

**Figure 5.** IRR histograms obtained from a 1000-sample Monte Carlo analysis applying random technology variations in the NMOS for (**a**) upconversion, (**b**) downconversion, (**c**) upconversion with a fixed 1% source impedance mismatch, and (**d**) downconversion with a fixed 5% load impedance mismatch.

Although a NMOS mismatch in upconversion seems to be more detrimental for a double quadrature scheme than for a simple quadrature scheme (as shown in Figure 3c), the histogram from Figure 5a reveals that the difference is notably reduced when a statistical analysis is carried out instead of the worst-mismatch scenario. In other words, both schemes present an almost equal IRR distribution from the Monte Carlo analysis.

Additionally, the histograms from downconversion present a similar distribution for both schemes (represented in Figure 5b), although, in this case, the double quadrature scheme shows greater robustness against worst-mismatch combination IRR degradation than the single quadrature scheme (see Figure 4c).

Furthermore, cases with a certain fixed deviation in IF impedances were also simulated. This allowed us to observe how impedance imbalances and mixer mismatches are combined. In Figure 5c,d, we included a dashed line to indicate the IRR without a transistor mismatch (considering only the IF impedance mismatch).

The upconversion case (Figure 5c) shows a similar distribution to that in Figure 5a but is shifted to a lower IRR because of the 1% IF mismatch. Without the transistor mismatch, the source deviation limits the IRR to 48 dB and 49 dB for the single and double quadrature schemes, respectively. It is noticeable how the single quadrature distribution becomes slightly worse than the double quadrature distribution, although without the IF mismatch it is otherwise. This difference is explained by the results of Figure 3b, shown in Figure 5c as a dashed line for the applied mismatch.

Figure 5d represents the downconversion case with a 5% IF mismatch. An IRR of 47 dB (single quadrature) and 56 dB (double quadrature) was achieved without any deviation in the transistors (only the effect of the IF mismatch). In it, the single quadrature and double quadrature schemes present a different range in their IRR distribution. This is caused by the different severity of the IF mismatch in each scheme (see Figure 4b). Note also that the stricter restriction of the fixed mismatch causes a reduction in variability (a narrower histogram).

## 4. Discussion

Two schemes (simple and double quadrature) were evaluated in terms of both upconversion and downconversion. Both architectures employ the same mixer design, and, hence, the double quadrature approach will require more area, although the additional components should improve the performance according to the literature. However, the quantitative benefits are not easy to estimate. For this reason, Table 2 summarizes this study to provide a clear comparison between both strategies.

**Table 2.** Analysis results. The percentages indicate the maximum admissible deviation to not decrease the IRR under the 40 dB IRR limit. Statistical results present the median value and 15% and 85% quartiles from the Monte Carlo analysis (see Figure 5).

| Analysis | | | Upconversion | | Downconversion | |
|---|---|---|---|---|---|---|
| | | | SQ | DQ | SQ | DQ |
| Input Quadrature | IF/RF | $\Delta\theta$ | 1.5% | 1.4% | - | - |
| | | $A_{BAL}$ | 2.7% | 3.2% | - | - |
| | LO | $\Delta\theta$ | 1.35% | - | 1.25% | - |
| | | $A_{BAL}$ | 2.9% | - | 2% | - |
| Impedance balance | IF | | 2.5% | 3% | * | * |
| | LO | | 2.5% | - | 3% | - |
| Device mismatch | Worst mismatch scenario | | 2% | 1.4% | 2.5% | * |
| | Monte Carlo analysis | 15% | 56.9 dB | 56 dB | 62.5 dB | 64.6 dB |
| | | median | 62.6 dB | 62.8 dB | 67.6 dB | 70.5 dB |
| | | 85% | 70.4 dB | 73.5 dB | 74.2 dB | 79.9 dB |

- indicates that the deviation does not alter the IRR; * marks cases whose deviation to degrade under the 40 dB IRR must be higher than the expected mismatch.

This study evaluates the impact of each effect separately and does not consider the dependencies between the effects other than $A_{BAL}$ and $\Delta\theta$ from the same signal. This means that double quadrature cancels LO errors for upconversion and, due to the downconversion image's nature, neither IF nor LO errors alone will affect the IRR at the first order of approximation. Thus, for this scheme, only IF quadrature results in IRR degradation with similar limits on the single quadrature scheme. The single quadrature architecture only neglects RF quadrature errors, as it is a differential signal. All of the remaining quadrature signals present similar constraints. However, they are noticeably strict for downconversion LO, making the double quadrature scheme especially attractive for conversion to IF.

The impedance balance is significantly more critical to upconversion than to downconversion. The former implies a 2.5% limit for an imbalance in IF or LO impedances in the single quadrature scheme to be able to reach the 40 dB IRR. The major improvement from the double quadrature scheme is to cancel the LO impedance error, although it slightly relaxes the IF impedance mismatch constraint to 3%. Downconversion restrictions are significantly looser by comparison: IF impedances can tolerate a mismatch higher than 10%, with the double quadrature approach attaining a better IRR. Again, the main difference between the performance of both schemes is LO robustness. The double quadrature circuit cancels the mismatch impact, while the simple approach requires less than a 3% LO impedance imbalance.

The worst mismatch scenario also presents stricter limits on upconversion than on downconversion. The improvement is small for the simple quadrature scheme but significant for the double quadrature scheme. Surprisingly, the simple quadrature approach presented better performance than the double quadrature one when the worst mismatch scenario was evaluated.

However, the probability of occurrence (the worst possible combination will be more common on four devices than eight) ameliorates this effect to point that, in the statistical analysis, the mean is better for the double quadrature scheme. In more extreme samples, the results approach the worst mismatch scenario, and, hence, the simple quadrature approach has a better IRR. An opposite but similar situation occured in the downconversion case: while the double quadrature scheme showed a huge difference (10 dB) in the worst mismatch scenario, it only had a roughly 3 dB IRR advantage in the statistical analysis.

To sum up, upconversion presents stricter limits in all but the quality of the LO signal quadrature (only for the simple scheme). However, this variation is small enough to not force the use of different schemes in upconversion and downconversion. The most significant advantage of double quadrature is relaxing LO-related parameters. Thus, the convenience of employing twice the area for the mixer stage strongly depends on the quality of the LO signal and the availability of silicon surface.

## 5. Conclusions

In this paper, an analysis of the IRR for CMOS passive mixers was carried out for upconversion and downconversion in simple and double quadrature schemes for 65 nm standard CMOS technology. The analysis studied the effects on IRR of errors in the quality of the input quadrature, impedance balance, and mismatch device (worst mismatch scenario and statistical analysis).

The results show that the double quadrature scheme provides better performance in almost all cases, and hence it usually will impose less strict restrictions on the design of contiguous stages and the mixer unit itself. However, using twice the area and a more complex design may not be worthwhile if the main concern is IF balance on upconversion, as there is only a minor improvement. The same happens if the LO signal's quality is not an issue.

The study also reveals that the mismatch constraints are as important in the mixers as they are in the previous and following blocks. Indeed, the mismatch between the quadrature branches on those circuits will be a critical factor in deciding whether a simple or double quadrature structure should be employed. While the benefit from a double quadra-

ture structure in the NMOS mismatch case is limited (see Monte Carlo analysis), the double quadrature scheme notably relaxes the constraints on the source/load impedance balance.

**Author Contributions:** Methodology, A.D.M.-P., F.A., and S.C.; analysis, A.D.M.-P.; writing—draft preparation, A.D.M.-P.; writing—review and editing, F.A., G.R., and S.C.; supervision, F.A. and S.C.; All authors have read and agreed to the published version of the manuscript.

**Funding:** This work was funded by the Spanish MINECO-FEDER project TEC2017-85867-R and a DGA PhD Scholarship to Antonio D. Martinez-Perez.

**Data Availability Statement:** Data is contained within the article.

**Conflicts of Interest:** The authors declare no conflict of interest.

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
