# Peer review of "Analysis of Non-Idealities on CMOS Passive Mixers"

_electronics, doi:10.3390/electronics10091105_

Round 1

Reviewer 1 Report

  1. In Abstract, the authors need to provide simulated or measured results and to highlight the contribution with any advanced methods or techniques.
  2. Figure 1 is a generic schematic of the MOS passive mixer, which is published in [4]. The authors need to draw the RF and IF terminals, not IN and OUT. Besides, Fig. (a) can be deleted because it makes no sense.
  3. In Introduction, those previous studies need to be cited and analyzed their advantages and disadvantages. Finally, the authors need to propose a novel method or technique to improve the performance of this manuscript.
  4. In line 111, page 4, the “Table 1. …” needs to be changed to the top of table, so does Table 2 in page 9.
  5. Please review the English style along with the entire paper. I have detected some typos (i.e. "…SQ and DQ have been divided for…" instead of "…SQ and DQ has been divide for…" in lines 145-146, page 4.)
  6. As show in Fig. 4(a), the IRR does not show with respect to LO phase error and amplitude imbalance.
  7. This manuscript studies the effects of device mismatches and quadrature signal errors on a conversion. However, there are many impacts in IRR. The authors need to consider others parameters, not limited to two parameters.
  8. More references must be added and a comparison needs to be completed to highlight the novelty of this study.
  9. Measurements are critical. Please give the measured results to prove the feasibility.

Reviewer 2 Report

An analysis of the IRR for passive mixers is presented using 65-nm CMOS technology. The motivation, analysis, and obtained results are well presented. I would suggest to the authors only extend the literature survey, especially for the claims in the introduction. 

Author Response

The authors would like to thank the reviewer for their comment and their positive appreciation of our work. The new version includes additional references to support the claims in the introduction and an analysis of previous studies. The changes are highlighted in red.

Reviewer 3 Report

The paper is interesting and well-written. The method is adequately described and well supported by the reported results. I recommented to accept it in the present form.

Author Response

The authors would like to thank the reviewer for their comment and their positive appreciation of our work.

Round 2

Reviewer 1 Report

  1. A comparison table needs to be completed to highlight the novelty of this study.
  2. Measurements are critical. Please give the measured results to prove the practicality.
  3. As an analysis of non-ideality mixer, more theoretical equations and adopted values need to be provided to let those readers know how to complete those simulated figures in this manuscript.
  4. If possible, the authors can add some flowcharts to comprehensive this simulation procedure.
